# VISUALIZING AND UNDERSTANDING GANS

**David Bau[1], Jun-Yan Zhu[1], Hendrik Strobelt[2], Bolei Zhou[3],**
**Joshua B. Tenenbaum[1], William T. Freeman[1], Antonio Torralba[1]**
[1]Massachusetts Institute of Technology, [2]IBM Research, Cambridge MA,
[3]The Chinese University of Hong Kong

The ability of generative adversarial networks to render nearly photorealistic images leads us to ask: What does a GAN know? For example, when a GAN generates a door on a building but not in a tree, we wish to understand whether such structure emerges as pure pixel patterns without explicit representation, or if the GAN contains internal variables that correspond to human-perceived objects such as doors, buildings, and trees. And when a GAN generates an unrealistic image, we want to know if the mistake is caused by specific variables in the network.

We first identify a group of interpretable units that are related to semantic classes (Figure 1a,b). These units' featuremaps closely match the semantic segmentation of a particular object class (e.g., trees). Then, we intervene in units in the network to cause a type of object to disappear or appear (Figure 1c,d). Finally, we study contextual relationships by observing where we can insert the object concepts in new images and how this intervention interacts with other objects in the image (Figure 4). This framework allows us to compare representations across different layers, GAN variants, and datasets; to debug and improve GANs by locating artifact-causing units (Figure 1e-g); to understand contextual relationships between objects in natural scenes (Figure 4, Figure 5); and to manipulate images with interactive object-level control (video).

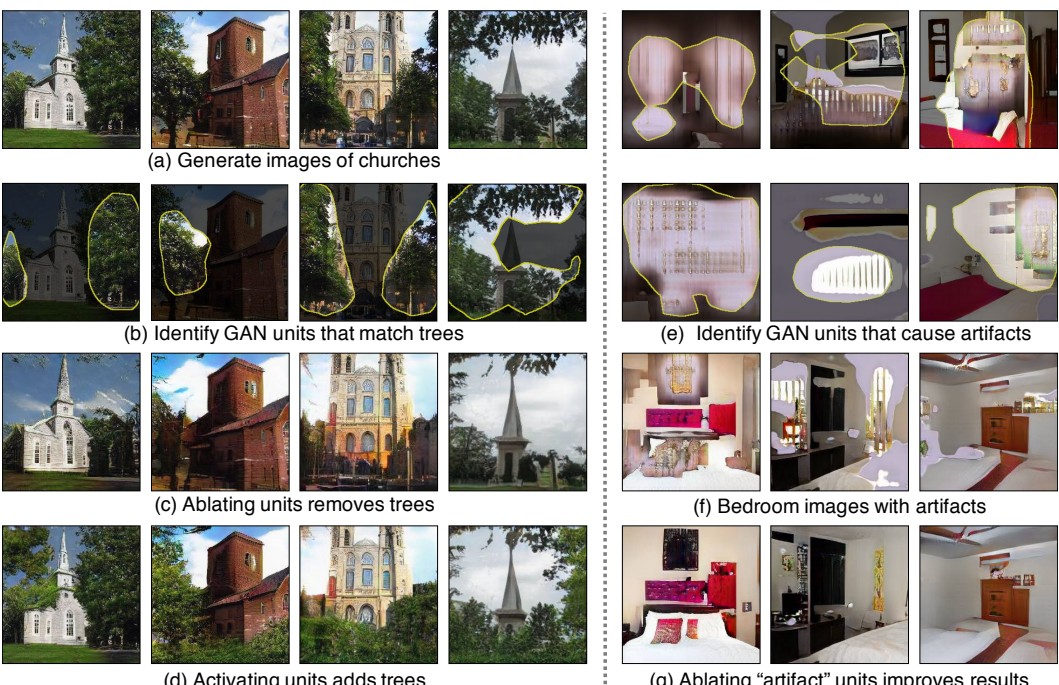

(a) Generate images of churches

(b) Identify GAN units that match trees

(c) Ablating units removes trees

(d) Activating units adds trees

(e) Identify GAN units that cause artifacts

(f) Bedroom images with artifacts

(g) Ablating "artifact" units improves results

Figure 1: Overview: (a-d) We analyze internal representations by relating (a) output of a GAN to (b) units that correlate with object concepts (e.g., trees) and intervening to (c) remove and (d) add objects. We can (e) identify units that (f) cause artifacts and (g) reduce artifacts when ablated.

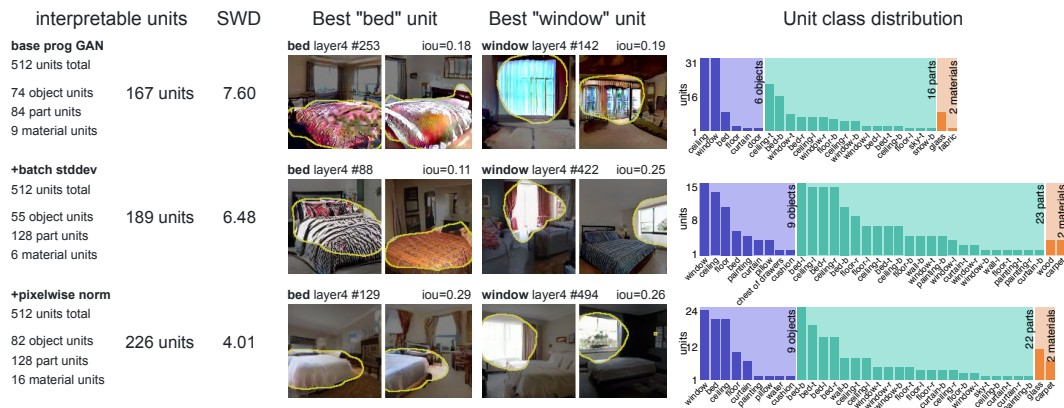

Figure 2: Comparing `layer4` representations learned by different training variations. Lower SWD indicates a higher-quality model: as the quality of the model improves, the number of interpretable units also rises. Progressive GANs apply several innovations including making the discriminator aware of minibatch statistics, and pixelwise normalization at each layer. We can see batch awareness increases the number of object classes matched by units, and pixel norm (applied in addition to batch stddev) increases the number of units matching objects.

## 1 METHOD

We analyze the internal GAN representations by decomposing the featuremap $\mathbf{r}$ at a layer into positions $P \subset \mathbb{P}$ and unit channels $u \in \mathbb{U}$. To identify a unit $u$ with semantic behavior, we upsample and threshold the unit, and measure how well it matches an object class $c$ in the image $\mathbf{x}$ as identified by a supervised semantic segmentation network $\mathbf{s}_c(\mathbf{x})$ (Xiao et al., 2018)

$$\text{IoU}_{u,c} \equiv \frac{\mathbb{E}_{\mathbf{z}} \left| (\mathbf{r}_{u,\mathbb{P}}^{\uparrow} > t_{u,c}) \wedge \mathbf{s}_c(\mathbf{x}) \right|}{\mathbb{E}_{\mathbf{z}} \left| (\mathbf{r}_{u,\mathbb{P}}^{\uparrow} > t_{u,c}) \vee \mathbf{s}_c(\mathbf{x}) \right|}, \qquad \text{where } t_{u,c} = \arg\max_t \frac{\text{I}(\mathbf{r}_{u,\mathbb{P}}^{\uparrow} > t; \mathbf{s}_c(\mathbf{x}))}{\text{H}(\mathbf{r}_{u,\mathbb{P}}^{\uparrow} > t, \mathbf{s}_c(\mathbf{x}))} \qquad (1)$$

This approach is inspired by the observation that many units in classification networks locate emergent object classes when upsampled and thresholded (Bau et al., 2017). Here, the threshold $t_{u,c}$ is chosen to maximize the information quality ratio, that is, the portion of the joint entropy H which is mutual information I (Wijaya et al., 2017).

To identify a sets of units $U \subset \mathbb{U}$ that cause semantic effects, we intervene in the network $G(\mathbf{z}) = f(h(\mathbf{z})) = f(\mathbf{r})$ by decomposing the featuremap $\mathbf{r}$ into two parts $(\mathbf{r}_{U,P}, \mathbf{r}_{\overline{U,P}})$, and forcing the components $\mathbf{r}_{U,P}$ on and off. Given an original image $\mathbf{x} = G(\mathbf{z}) \equiv f(\mathbf{r}) \equiv f(\mathbf{r}_{U,P}, \mathbf{r}_{\overline{U,P}})$, we can intervene in the network and generate an image with units U ablated at pixels P:

$$\mathbf{x}_a = f(\mathbf{0}, \mathbf{r}_{\overline{U,P}}) \qquad (2)$$

Or an image with units U activated to a high level $\mathbf{c}$ at pixels P:

$$\mathbf{x}_i = f(\mathbf{c}, \mathbf{r}_{\overline{U,P}}) \qquad (3)$$

We measure the average causal effect (ACE) (Holland, 1988) of units U on class $c$ as:

$$\delta_{U \to c} \equiv \mathbb{E}_{\mathbf{z},P}[\mathbf{s}_c(\mathbf{x}_i)] - \mathbb{E}_{\mathbf{z},P}[\mathbf{s}_c(\mathbf{x}_a)], \qquad (4)$$

## 2 RESULTS AND DISCUSSION

Analysis of the semantics and causal behavior of the internal units of a GAN reveals several new findings.

**Units matching diverse objeccts emerge on more diverse models.** Internal units for more object classes emerge as the architecture becomes more diverse. Figure 2 compares three models (Karras et al., 2018) that introduce two innovations on baseline Progressive GANs. The number of types of objects, parts, and materials matching units increases by more than 40% as minibatch-stdev is introduced; and pixelwise normalization increase units that match semantic classes by 19%.

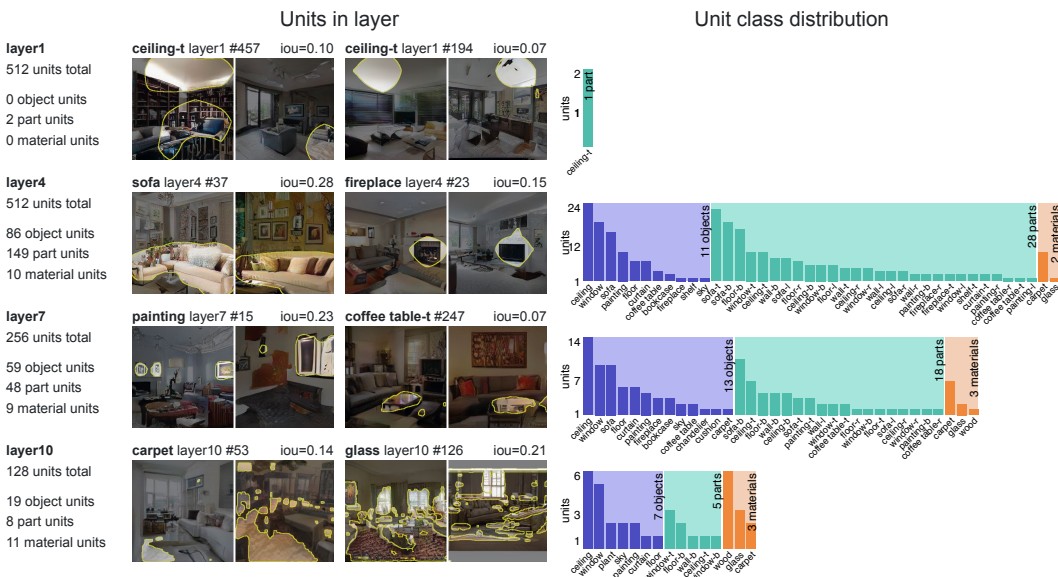

Figure 3: Comparing layers of a progressive GAN trained to generate $256 \times 256$ LSUN living room images. The output of the first convolutional layer has almost no units that match semantic objects, but many objects emerge at layers 4-7. Later layers are dominated by low-level materials and shapes.

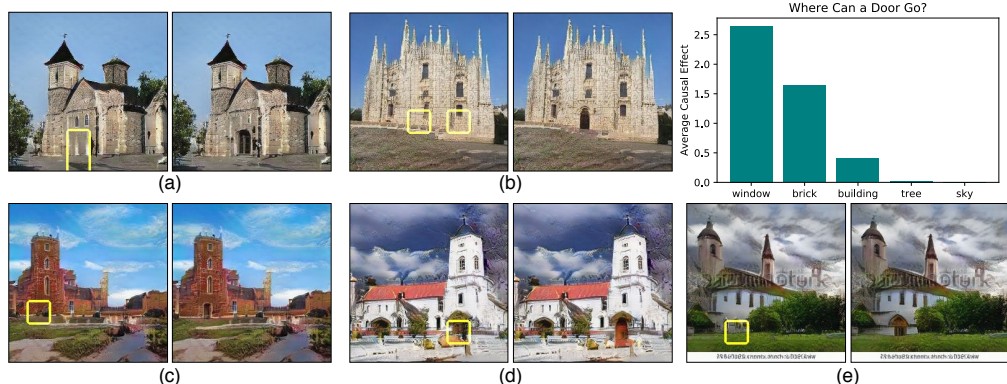

Figure 4: Inserting door units by setting 20 causal units to a fixed high value at one pixel in the representation. Whether the door units can cause the generation of doors is dependent on local context: every location that creates doors is shown, including two separate locations in (b) (we intervene at left). The same units are inserted in every case, but the door that appears has a size, alignment, and color appropriate to the location. The top chart summarizes the causal effect of inserting door units at one pixel with different context.

**Interpretable units emerge in the middle layers, not at the initial layers.** In classifier networks, units matching high-level concepts appear in layers furthest from the pixels (Zeiler & Fergus, 2014). In contrast, in a GAN, it is mid-level layers 4 to 7 that have the largest number of units that match semantic objects and object parts. A selection of layers is shown in Figure 3.

**Diagnosing and Improving GANs** Our framework can also analyze the causes of failures and repair some GAN artifacts. Figure 1e shows several annotated units that are responsible for typical artifacts consistently appearing across different images. We can fix these errors by ablating 20 artifact-causing units. Figure 1g shows that artifacts are successfully removed and the artifact-free pixels stay the same, improving the generated results. Table 1 summarizes quality improvements: we compute the Fréchet Inception Distance (Heusel et al., 2017) between the generated images and real images using 50 000 real images and 10 000 generated images with high activations on these units. We also collect 20 000 annotations of realism on Amazon MTurk, with 1 000 images per method.

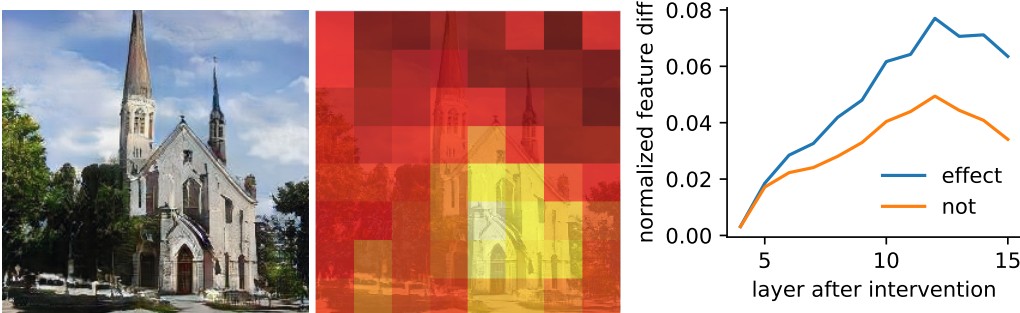

Figure 5: Tracing the effect of inserting door units on downstream layers. An identical "door" intervention at `layer4` of each pixel in the featuremap has a different effect on final convolutional feature layer, depending on the location of the intervention. In the heatmap, brighter colors indicate a stronger effect on the `layer14` feature. A request for a door has a larger effect in locations of a building, and a smaller effect near trees and sky. At right, the magnitude of feature effects at every layer is shown, measured by mean normalized feature changes. In the line plot, feature changes for interventions that result in human-visible changes are separated from interventions that do not result in noticeable changes in the output.

Table 1: We compare generated images before and after ablating 20 "artifacts" units. We also report a simple baseline that ablates 20 randomly chosen units.

| Fréchet Inception Distance (FID) | |
| --- | --- |
| original images | 52.87 |
| "artifacts" units ablated (ours) | **32.11** |
| random units ablated | 52.27 |

| Human preference score | original images |
| --- | --- |
| "artifacts" units ablated (ours) | **79.0**% |
| random units ablated | 50.8% |

**Characterizing contextual relationships using insertion**   We can also learn about the operation of a GAN by forcing units on and inserting these features into specific locations in scenes. Figure 4 shows the effect of inserting 20 `layer4` causal door units in church scenes. We insert units by setting their activation to the mean activation level at locations at which doors are present. Although this intervention is the same in each case, the effects vary widely depending on the context. The doors added to the five buildings in Figure 4 appear with a diversity of visual attributes, each with an orientation, size, material, and style that matches the building. We also observe that doors cannot be added in most locations. The locations where a door can be added are highlighted by a yellow box. The bar chart in Figure 4 shows average causal effects of insertions of door units, conditioned on the object class at the location of the intervention. Doors can be created in buildings, but not in trees or in the sky. A particularly good location for inserting a door is one where there is already a window.

**Tracing the causal effects of an intervention**   To investigate the mechanism for suppressing the visible effects of some interventions, we perform an insertion of 20 door-causal units on a sample of locations and measure the changes in later layer featuremaps caused by interventions at layer 4. To quantify effects on downstream features, and the effect on each each feature channel is normalized by its mean L1 magnitude, and we examine the mean change in these normalized featuremaps at each layer. In Figure 5, these effects that propagate to `layer14` are visualized as a heatmap: brighter colors indicate a stronger effect on the final feature layer when the door intervention is in the neighborhood of a building instead of trees or sky. Furthermore, we graph the average effect on every layer at right in Figure 5, separating interventions that have a visible effect from those that do not. A small identical intervention at `layer4` is amplified to larger changes up to a peak at `layer12`.

Interventions provide insight on how a GAN enforces relationships between objects. We find that even if we try to add a door in `layer4`, that choice can be vetoed by later layers if the object is not appropriate for the context.

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
