# OpenReview forum: "Visualizing and Understanding GANs"
_ICLR.cc/2019/Workshop/DeepGenStruct — DeepGenStruct 2019_

### Official Review · AnonReviewer1 · 2019-04-12
**Interesting topic and carefully-designed experiments**

**Rating:** 4
**Confidence:** 2

**Review:**

This paper tries to ask and answer a challenging question: What does a GAN know? This is an interesting topic, and the authors proposed a simple method that can be used to visualize and understand the internal representations that GAN has learned.

Pros: The experimental results are very interesting. For example, by using the proposed approach, one can identify GAN units that match trees, and therefore one can ablate the corresponding units to remove tree, or activate the corresponding units to add tree. The authors also provide many other quantitative results in order to better understand GAN. The provided video link is also very interesting. It provides a kind of controllable manner to manipulate generated images.

 Cons: Though the proposed method looks simple, I find it a little bit difficult for me to follow the details. More clarity is needed.

---

### Official Review · AnonReviewer2 · 2019-04-17
**nice work in general**

**Rating:** 4
**Confidence:** 2

**Review:**

This work describes a simple method to visualize and understanding GANS. The method starts with identifying semantic object classes in an image (with a supervised semantic segmentation network) and then manipulates the network units to study the impact. It shows cool applications of object-level control of images.

My main concern of this approach is I'm not sure if the method is image-specific or not. If it is, that means for the same object occurring in different images, there will be different network units responsible. If this is the case there wouldn't be much conclusion which can be drawn from the results.

---

### Decision · Program_Chairs · 2019-04-19
**Acceptance Decision**

Accept